# Exploring the Potential of Tomato Processing Byproduct as a Natural Antioxidant in Reformulated Nitrite-Free Sausages

**Andreea I. Cadariu** [1], **Ileana Cocan** [1,*] , **Monica Negrea** [1] , **Ersilia Alexa** [1] , **Diana Obistioiu** [2] , **Ionela Hotea** [2] , **Isidora Radulov** [3] and **Mariana-Atena Poiana** [1,*]

1   Faculty of Food Engineering, University of Life Sciences "King Michael I" from Timisoara, Calea Aradului 119, 300645 Timisoara, Romania
2   Faculty of Veterinary Medicine, University of Life Sciences "King Michael I" from Timisoara, Calea Aradului 119, 300645 Timisoara, Romania
3   Faculty of Agriculture, University of Life Sciences "King Michael I" from Timisoara, Calea Aradului 119, 300645 Timisoara, Romania
*   Correspondence: ileanacocan@usab-tm.ro (I.C.); marianapoiana@usab-tm.ro (M.-A.P.);
    Tel.: +40-765-265-882 (I.C.); +40-726-239-838 (M.-A.P.)

**Abstract:** This study evaluated the potential of two dried processing by products, obtained from large and cherry tomatoes (LT and CT) after juice extraction to improve the oxidative stability of pork sausages during refrigerated storage for 20 days. For this purpose, reformulated nitrite-free sausages were manufactured by supplementation of raw sausage samples with dried large and cherry tomato processing byproducts (DLTB and DCTB) at a dose that provides a level of polyphenolic compounds equally with 50, 90, 180 and 270 mg gallic acid equivalents (GAE)/kg of processed meat. The developed sausage formulas were subjected to heat treatment, such as smoking and drying and, smoking and scalding, respectively. The reformulated nitrite-free sausages were compared with control samples of sausages with or without the addition of sodium nitrite. Large and cherry tomatoes and their raw and dried processing byproducts were investigated for total and individual polyphenols content as well as lycopene content. The sausage formulas were evaluated in terms of proximate composition. Additionally, the progress of lipid oxidation developed in sausage formulas was assessed by specific indices, such as peroxide value (PV), *p*-anisidine value (*p*-AV), TOTOX value and thiobarbituric acid reactive substances (TBA), after 1, 10 and 20 days of storage at 4 °C. Based on the values of PV, *p*-AV, TOTOX and TBA, it can be stated that the dried tomato processing byproducts applied at doses that ensure a level of polyphenolic compounds of at least 180 mg GAE/kg of processed meat for DCTB and 270 mg GAE/kg of processed meat for DLTB, showed promising potential to replace sodium nitrite in meat products for both dried and scalded sausage formulas. For the same dose of tomato processing byproducts, it was noted a stronger inhibitory effect against lipid oxidation in the case of smoked and scalded sausages compared to smoked and dried ones.

**Keywords:** tomato processing byproducts; nitrite-free sausages; lipid oxidation; natural antioxidant

## 1. Introduction

Sausages are a category of meat preparations widely consumed in many countries and are formed from a mixture of proteins and fats from meat with additives [1].

Recently, the concerns for a healthy diet have led to the need to reformulate the classic recipes for the manufacture of pork meat products in order to obtain beneficial health properties. Regarding meat products, considerable efforts are being made to reformulate them by adding functional ingredients.

Lipid oxidation represents the primary factor for meat products' quality to decline during processing and storage. The chemical compounds resulting from primary and secondary oxidation not only change the flavor, color and texture of meat products, but also reduce their nutritional quality.

The composition of the ingredients used, the temperature and duration of the heat treatment applied and the rate of grinding and emulsification influence the properties of finished products [2]. However, proteins and lipids in meat are affected by continuous exposure to the oxidizing environment [3].

The interaction of free acids and oxygen in the presence of heat and the chemical composition and processing technique can cause meat products to oxidize [4]. While protein oxidation, which is always present alongside lipid oxidation, affects sausage quality in terms of tastes, texture and biological functioning, and excessive lipid oxidation damages the quality and imparts an off-flavor [5]. Meanwhile, new research claims that excessive use of synthetic food additives is always associated with gastrointestinal difficulties, respiratory diseases, dermatological concerns and harmful neurological effects [6]. Oxidation processes might degrade the nutritional value, functional impacts, and sensory characteristics that are a quality feature in muscle meals, lowering their shelf life [7].

Applying an antioxidant to meat products can minimize lipid oxidation during storage, extending their shelf life and maintaining their quality and safety [8]. In order to prevent food from spoiling, a variety of synthetic preservatives, including sodium nitrate, sodium nitrite, butylated hydroxyanisole (BHA), butylated hydroxytoluene (BHT) and propyl gallate (PG), are frequently used. However, their use is restricted due to the possibility that they may have cancer-causing effects. As a result, increased demand for natural alternatives to chemicals has gradually sparked a movement to exclude synthetic preservatives from food [9].

Numerous investigations are being undertaken to find natural additives with broad antioxidant activity to enhance meat products' quality and shelf life without compromising their organoleptic features [10]. Sausages have frequently been preserved using sodium nitrite/nitrate. Additionally, nitrite and nitrate salts are added to meat products to provide the distinctive pink color, flavor and fragrance of cured cooked goods. They also help to postpone lipid oxidation by chelating metal ions [11]. As one of the most popularly cultivated and consumed vegetables, tomatoes are a rich source of lycopene, beta-carotene, folate, potassium, chlorogenic acid, flavonoids, plastoquinone, phenols, tocopherol (vitamin E) and xanthophylls [12–14]. Lycopene is a crucial antioxidant that is very stable during storage and cooking. As a result, it may be found in cooked tomatoes, which are widely eaten and help reduce the risk of heart disease and cancer [13,15]. Additionally, many epidemiological studies have hypothesized that frequent tomato consumption may be associated with lower rates of cardiovascular disease [16] and a lower risk of breast, colon, lung and prostate cancer [17].

Although previous research attests that tomato processing byproducts are a good source of bioactive chemicals and dietary fiber, they are underutilized in the canned food sector [15]. It is impossible to completely eliminate the significant waste produced by the vegetable processing sector and using processing waste in a more sustainable way can lead to reduced negative effects on the environment [18].

Thus, this paper aimed to investigate the potential of tomato processing byproduct as natural additive for smoked and dried and smoked and scalded sausages, respectively. Our hypothesis is that the bioactive compounds present in tomato byproducts can replace the synthetic additives used in meat products, reducing the costs and environmental problems generated by the removal of these residues. In the light of these mentions, this work aimed to evaluate the effect of tomato processing byproduct as a natural additive for improving the oxidative quality and extending the shelf life of meat products. Our theory is that the bioactive compounds found in tomato byproducts can replace synthetic additives used in meat products, lowering prices and preventing the environmental issues caused by these residues. This study aimed to assess how tomato byproducts could be used as a natural additive to enhance meat products' stability to oxidation and lengthen their shelf lives. The proximate composition and oxidative stability were evaluated for the designed sausage formulas.

## 2. Materials and Methods

### 2.1. Materials

Fresh meat and fat were purchased from local processors (SC Smithfield SRL, Romania). Fresh tomatoes (*Solanum lycopersicum*) and garlic were purchased from local supermarkets (Timisoara, Timis County, Romania). Two types of tomatoes (cherry tomatoes and large tomatoes) were used. Ingredients such as ground black pepper (Fuchs Condimente RO SRL, Romania), sweet paprika (Fuchs Condimente RO SRL, Romania) and salt (Salrom, Romania) were purchased from a local market. All reagents used in chemical analyses were of analytical grade and were purchased from the companies Sigma-Aldrich (St. Louis, MO, USA) and Chimreactiv (Bucharest, Romania).

### 2.2. Obtaining the Tomatoes Processing Byproducts

Two types of tomatoes (cherry and large tomatoes) were processed for juice obtaining, and the raw resulting byproducts were collected and conditioned by drying into a forced air oven (Froilabo AC60/France, 1000 W) at 60 °C for 16 h to avoid the bioactive compounds' degradation. After drying, the tomato byproduct samples were ground with a laboratory mill Grindomix GM 2000 (Retsch GmbH, Germany) until they were turned into a fine powder that was passed through a 60-mesh sieve. This powder was further incorporated in the sausage recipe. The following abbreviations are used for the fresh investigated samples: CT—cherry tomatoes; LT—large tomatoes; CTB—cherry tomato byproduct; LTB—large tomato byproduct. Similarly, for the resulting byproduct samples after drying, the following abbreviations are used: DCTB—dried cherry tomato byproduct; DLTB—dried large tomato byproduct.

### 2.3. Phytochemical Profile of Tomatoes and Tomato Processing Byproducts

#### 2.3.1. Preparation of the Alcoholic Extracts

Each sample (1 g) of large and cherry tomatoes as well as their raw and dried processing byproducts was weighed in containers with lids and then 10 mL of 70% (*v/v*) ethanol (Chimreactiv, Bucharest, Romania) was added. The containers were closed with lids and stirred for 30 min using a magnetic stirrer (IDL, Freising, Germany). The samples were stirred before filtering through Whatman N°1 filter paper. The extracts thus obtained were further used to determine the content of total and individual polyphenols as well as lycopene content.

#### 2.3.2. Assessment of Total Phenolic Content (TPC)

The total content of polyphenols (TPC) in tomatoes and their raw and dried processing byproducts was determined according to the modified Folin–Ciocalteu method [19]. A 0.5 mL aliquot of each extract was taken and 1.25 mL Folin–Ciocalteu reagent (Sigma-Aldrich Chemie GmbH, Munich, Germany) diluted to 1:10 (*v/v*) with demineralized water was added. The samples were treated with 1 mL of 60 g/L $Na_2CO_3$ (Geyer GmbH, Renningen, Germany) aqueous solution after standing for 5 min at room temperature. The absorbance of the samples was measured at 750 nm using the UV-VIS spectrophotometer Specord 205, Analytik Jena Inc. (Jena, Germany) after 30 min of incubation at 50 °C in the thermostat (INB500, Memmert GmbH, Schwabach, Germany) against a blank sample prepared in the same conditions. A calibration curve was made using gallic acid solutions (Fluka, Madrid, Spain) with different concentrations in the range 20–200 mg/L. The values of TPC were expressed as mg gallic acid equivalents (GAE)/g of dry substance (d.s). The results were reported as the mean value of three independent analyses $\pm$ standard deviation (SD).

#### 2.3.3. Chromatographic Determination of Non-Anthocyanin Polyphenols by LC-MS

Non-anthocyanin polyphenols of tomatoes and their raw and dried processing byproducts from previously obtained extracts were separated and identified using the liquid chromatography-mass spectrometry method (LC-MS). According to Shimadzu, Kyoto,

Japan, a Shimadzu LCMS-2010 EV system with electrospray ionization (ESI) was used [20]. The chromatographic system consists of an HPLC unit with an MS-2010 mass spectrometer connected in-line, a degasser, an auto sampler and solvent delivery pumps (LC-10AD). On a NUCLEODUR CE 150/2 C18 Gravity SB 150 mm column of 2.0 mm, particle size 5 m and running at 20 °C with a flow rate of 0.2 mL/min (Macherey-Nagel GmbH & Co. KG, Germany), the inverted phase was separated. The compounds were separated with the elution of gradients A (aqueous formic acid, pH = 3) and B (acetonitrile and formic acid solution, pH = 3). The gradient program was as follows: 5% B (0.01–20 min), 5–40% B (20.01–50 min 10 min), 40–95% B (50–55 min) and 95% B (55–60 min). The injection had a 20 L volume. The detector was set to a purchase range between 200 nm and 700 nm, and monitoring was done at 280 and 320 nm. Further, 1.25 scans/s served as the spectral acquisition rate (peak width: 0.2 min). With Shimadzu's Solution software, data were collected and cutting-edge integrated, and calibrations were carried out. The calibration curves were run in the range of 20–50 g/mL. The individual polyphenolic compounds detected in samples were expressed as mg/g dry substance (d.s). All determinations were performed in triplicate and the results were given as mean value ± standard deviation (SD).

2.3.4. Spectrophotometric Determination of Lycopene

Lycopene is extracted into hydrophobic solvents and measured using a spectrophotometer at 502 nm [21]. Using a combination of hexane, ethanol, and acetone (2:1:1) ($v/v/v$), Sharma and Le Maguer's approach was used to extract lycopene from tomatoes and their processing byproducts [21]. As a result, 25 mL of hexane, ethanol, and acetone solution was combined with 1 g of the homogenized tomato sample before being forcefully agitated with a mechanical shaker for 30 min. Then, 10 mL of distilled water is added, and the mixture is stirred continuously for an additional 2 min. Hexane was used as a blank sample, and the absorbance was determined using the UV-VIS spectrophotometer Specord 205, Analytik Jena Inc. (Jena, Germany) at a wavelength of 502 nm. The values were expressed in mg lycopene/kg dry substance (d.s). The results were reported as the mean value of three independent analyses ± standard deviation (SD).

*2.4. Manufacture of Sausages Formulas*

The powder obtained from each dehydrated tomato processing byproduct was used in the sausages recipe at four levels of concentration in order to ensure a level of polyphenolic compounds equally with 50, 90, 170 and 270 mg gallic acid equivalents (GAE)/kg raw processed meat. The dose of polyphenolic compounds coming from the tomato processing byproduct was chosen taking into account the minimum nitrites content added to a kg of raw processed meat (90 mg nitrites/kg processed meat).

The control and nitrite-free sausage formulas with the addition of tomato byproducts were prepared from the same batch of meat and fat. The basic recipe of sausages was as follows: pork meat 800 g, pork fat 200 g, salt 18 g, sweet paprika 6 g, garlic 16 g, white pepper 2 g and black pepper 2 g. In the positive control samples, the salt was replaced with a mixture of salt containing 0.5% sodium nitrite. Another eight samples of sausages were prepared using the basic recipe, in which nitrite was replaced by DCTB and DLTB in doses providing a level of polyphenolic compounds of 50, 90, 180 and 270 mg GAE/kg of raw processed meat. The production process for all sausage formulas is illustrated in Figure 1. Both control and sausage samples supplemented with DCTB and DLTB were further divided into two batches, one of which was subjected to smoking and drying treatments, and the samples from the second batch were processed by smoking and scalded, resulting in 21 sausage formulas, as follows:

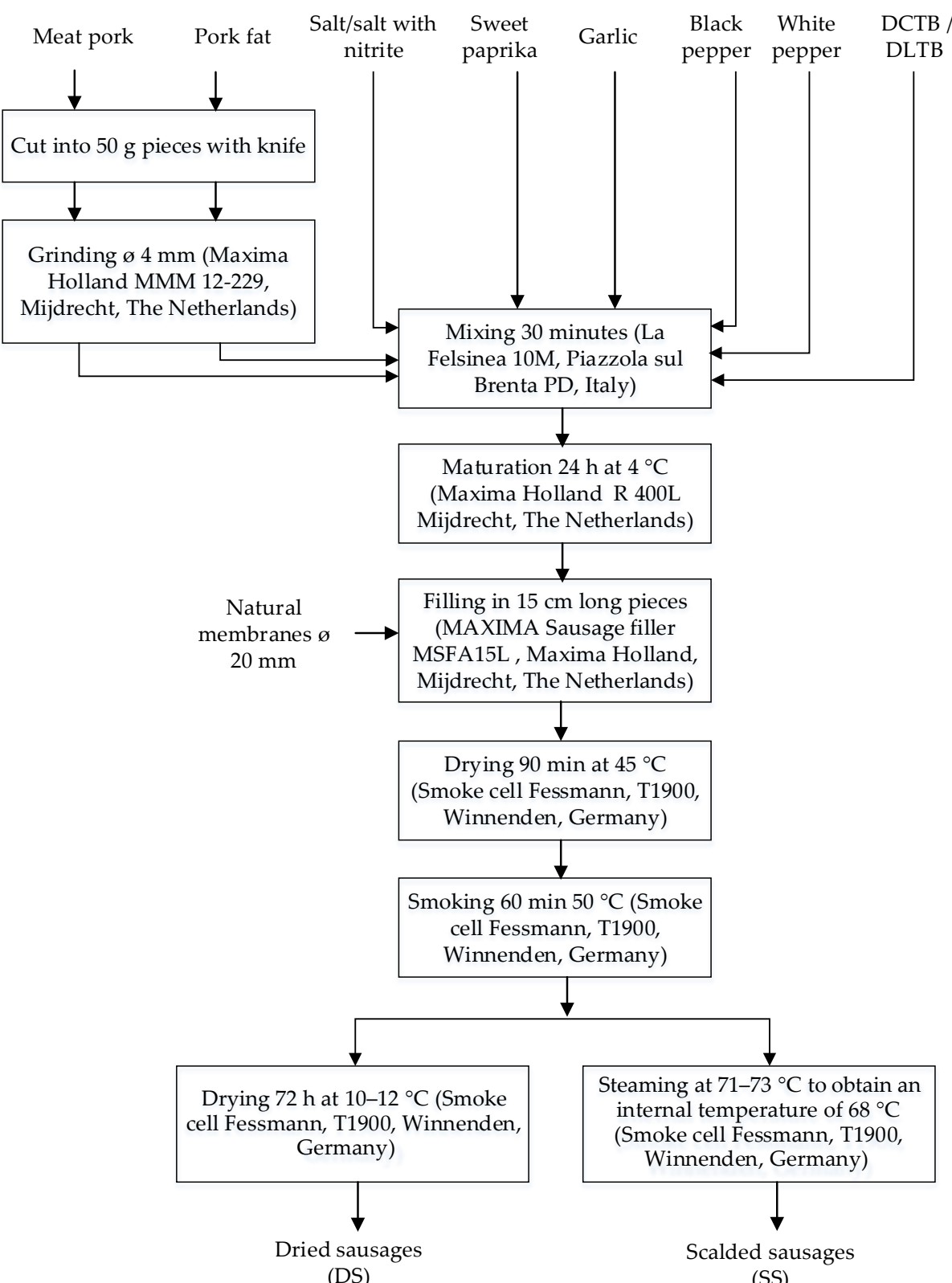

**Figure 1.** Technological flow diagram followed for obtaining the sausage formulas.

SC (sausages control): fresh sausages obtained according to the basic recipe;
SDC (dried sausages control): smoked and dried sausages obtained according to the basic recipe;

SDCN (dried sausages control with sodium nitrite): smoked and dried sausages with salt mixture containing 0.5% (*w/w*) sodium nitrite;

SSC (scalded sausages control): smoked and scalded sausages obtained according to the basic recipe;

SSCN (scalded sausages control with sodium nitrite): smoked and boiled sausages with salt mixture containing 0.5% (*w/w*) sodium nitrite;

DSDLTB (50, 90, 180, 270) (dried sausages with dried large tomato byproducts): smoked and dried sausages with different doses of DLTB that ensure a level of polyphenolic compounds of 50, 90, 180 and 270 mg GAE/kg of raw processed meat;

SSDLTB (50, 90, 180, 270) (scalded sausages with dried large tomato byproduct): smoked and scalded sausages with different doses of DLTB that ensure a level of polyphenolic compounds of 50, 90, 180 and 270 mg GAE/kg of raw processed meat;

DSDCTB (50, 90, 180, 270) (dried sausages with dried cherry tomato byproduct): smoked and dried sausages with different doses of DCTB that ensure a level of polyphenolic compounds of 50, 90, 180 and 270 mg GAE/kg of raw processed meat;

SSDCTB (50, 90, 180, 270) (scalded sausages with cherry tomato byproduct): smoked and scalded sausages with different doses of DCTB that ensure a level of polyphenolic compounds of 50, 90, 180 and 270 mg GAE/kg of raw processed meat.

The amounts of ingredients used to produce the 21 samples of sausages are given in Table 1. All amounts of ingredients were calculated according to the amount of meat and fat used.

**Table 1.** Amounts of ingredients used in the manufacturing recipe of sausage formulas.

| Sample | Pork Meat (g) | Pork Fat (g) | Salt (g) | Salt + 0.5% (*w/w*) Sodium Nitrite (g) | Sweet Paprika (g) | Garlic (g) | White Pepper (g) | Black Pepper (g) | DLTB (g) | DCTB (g) |
|---|---|---|---|---|---|---|---|---|---|---|
| SC | 800 | 200 | 18 | - | 6 | 16 | 2 | 2 | - | - |
| SDC | 800 | 200 | 18 | - | 6 | 16 | 2 | 2 | - | - |
| SDCN | 800 | 200 | - | 18 | 6 | 16 | 2 | 2 | - | - |
| SSC | 800 | 200 | 18 | - | 6 | 16 | 2 | 2 | - | - |
| SSCN | 800 | 200 | - | 18 | 6 | 16 | 2 | 2 | - | - |
| DSDLTB50 | 800 | 200 | 18 | - | 6 | 16 | 2 | 2 | 9.527 | - |
| DSDLTB90 | 800 | 200 | 18 | - | 6 | 16 | 2 | 2 | 17.149 | - |
| DSDLTB180 | 800 | 200 | 18 | - | 6 | 16 | 2 | 2 | 34.299 | - |
| DSDLTB270 | 800 | 200 | 18 | - | 6 | 16 | 2 | 2 | 51.448 | - |
| SSDLTB50 | 800 | 200 | 18 | - | 6 | 16 | 2 | 2 | 9.527 | - |
| SSDLTB90 | 800 | 200 | 18 | - | 6 | 16 | 2 | 2 | 17.149 | - |
| SSDLTB180 | 800 | 200 | 18 | - | 6 | 16 | 2 | 2 | 34.299 | - |
| SSDLTB270 | 800 | 200 | 18 | - | 6 | 16 | 2 | 2 | 51.448 | - |
| DSDCTB50 | 800 | 200 | 18 | - | 6 | 16 | 2 | 2 | - | 8.731 |
| DSDCTB90 | 800 | 200 | 18 | - | 6 | 16 | 2 | 2 | - | 15.715 |
| DSDCTB180 | 800 | 200 | 18 | - | 6 | 16 | 2 | 2 | - | 31.430 |
| DSDCTB270 | 800 | 200 | 18 | - | 6 | 16 | 2 | 2 | - | 47.145 |
| SSDCTB50 | 800 | 200 | 18 | - | 6 | 16 | 2 | 2 | - | 8.731 |
| SSDCTB90 | 800 | 200 | 18 | - | 6 | 16 | 2 | 2 | - | 15.715 |
| SSDCTB180 | 800 | 200 | 18 | - | 6 | 16 | 2 | 2 | - | 31.430 |
| SSDCTB270 | 800 | 200 | 18 | - | 6 | 16 | 2 | 2 | - | 47.145 |

Following preparation, the sausages were vacuum-packed in LD-PE bags using a vacuum-packaging machine (VAC-20 SL 2A, Edesa, Barcelona, Spain) and kept at 2 °C in the dark for 20 days.

The obtained sausage formulas were analyzed for the proximate composition and oxidative stability. Analyses to assess oxidative stability were performed after 1, 10 and 20 days of storage. The shelf life of smoked, dried and scalded meat products is 15 days. We chose in the present study to follow the oxidation degree for 20 days to see the evolution of the oxidation process after exceeding the shelf life. All analyses were carried out in triplicate for each category and each storage life.

Table 2 shows a clear view of the types of thermal treatment that were applied for each sausage formula designed in this study.

**Table 2.** Thermal treatments applied to sausage samples.

| Sample | Smoking | Drying | Scalding |
|---|---|---|---|
| SC | - | - | - |
| SDC | + | + | - |
| SDCN | + | + | - |
| SSC | + | - | + |
| SSCN | + | - | + |
| DSDLTB50 | + | + | - |
| DSDLTB90 | + | + | - |
| DSDLTB180 | + | + | - |
| DSDLTB270 | + | + | - |
| SSDLTB50 | + | - | + |
| SSDLTB90 | + | - | + |
| SSDLTB180 | + | - | + |
| SSDLTB270 | + | - | + |
| DSDCTB50 | + | + | - |
| DSDCTB90 | + | + | - |
| DSDCTB180 | + | + | - |
| DSDCTB270 | + | + | - |
| SSDCTB50 | + | - | + |
| SSDCTB90 | + | - | + |
| SSDCTB180 | + | - | + |
| SSDCTB270 | + | - | + |

(+) indicate the application of the treatment; (-) indicates the non-application of the treatment.

*2.5. Proximate Composition of Sausages*

The proximate composition of sausages was evaluated using the following ISO Methods: NaCl SR ISO 91-2007 [22], total lipid SR ISO 1443:2008 [23], total protein SR ISO 937:2007 [24], mineral substances SR ISO 936:2009 [25] and moisture SR ISO 1442:2010 [26].

The carbohydrates content of samples was determined based on the relationship shown in Equation (1), by calculating the percentage that remains after removing all other components that are measured, such as proteins, lipids, ash and moisture.

$$\text{Carbohydrates } (\%) = 100 - [\text{lipids}(\%) + \text{proteins}(\%) + \text{ash}(\%) + \text{moisture}(\%)] \quad (1)$$

The energy value of the items was calculated according to the relationship shown in Equation (2), by adding the caloric intake that each nutrient (lipids, carbohydrates and proteins) produced, considering that 1 g of carbohydrates provides 4 kcal, 1 g of proteins provides 4 kcal and 1 g of fats provides 9 kcal.

$$\text{Energy value } (\text{kcal}/100 \text{ g}) = \text{lipids } (\%) \times 9 + \text{proteins } (\%) \times 4 + \text{carbohydrates}(\%) \times 4 \quad (2)$$

*2.6. Oxidative Stability Assessment*

For subsequent determinations of the peroxide value (PV) and *p*-anisidine value (*p*-AV), from the investigated sausages samples, the fat was extracted with a Soxtest equipment (SX-6, Raypa Espinar, Terrassa, Barcelona, Spain), using as extraction solvent the petroleum

ether (Chimreactiv, Bucharest, Romania). Regarding the thiobarbituric acid (TBA) test, the analysis was performed directly on the sausage samples.

### 2.6.1. Determination of Peroxide Value (PV)

The PV of samples was evaluated using the iodometric method and the results are expressed as milliequivalents of active oxygen per kilogram (meq $O_2$/kg) of fat [27].

### 2.6.2. Determination of *p*-Anisidine Value (*p*-AV)

The *p*-anisidine value was evaluated based on the official spectrophotometric method (AOCS Official Method Cd 18-90) [28] by absorbance reading at 350 nm.

For this purpose, 2 g of sample was dissolved in 25 mL isooctane (Sigma-Aldrich Chemie GmbH, München, Germany) and then the absorbance was measured at 350 nm against a blank sample consisting of isooctane using the UV-VIS spectrophotometer Specord 205, Analytik Jena Inc. (Jena, Germany). Next, a 5 mL aliquot of the obtained solution of isooctane was separately transferred in two test tubes and, then, 1 mL of 0.25% (*w/v*) *p*-anisidine solution prepared in glacial acetic acid (Sigma-Aldrich Chemie GmbH, München, Germany) was added. The absorbance of the solution coming from the first test tube was read at 350 nm after 10 min against the solution consisting of isooctane with *p*-anisidine.

The *p*-AV was calculated using the relationship given in Equation (3):

$$p\text{-AV} = 25 \times \frac{1.2 \times A_2 - A_1}{W} \tag{3}$$

where $A_1$ is the the absorbance of fat sample in isooctane, $A_2$ is the the absorbance of fat sample in isooctane with *p*-anisidine solution and W is the fat sample weight (g).

### 2.6.3. Total Oxidation Value (TOTOX)

The relationship presented in Equation (4) was used to convert peroxide value (PV) and *p*-anisidine value (*p*-AV) into the TOTOX value [29].

$$\text{TOTOX} = 2 \times \text{PV} + p\text{-AV} \tag{4}$$

### 2.6.4. Thiobarbituric Acid (TBA) Test

The TBA test was performed following the technique described by Tran et al. [30] with a few minor adjustments. This method is based on the reaction of malondialdehyde (MDA) with thiobarbituric acid (TBA) and was specifically used to assess the lipids secondary oxidation developed in the sausage samples. Using a magnetic stirrer (IDL, Freising, Germany), 5 g of sausage was mixed with 20 mL of trichloroacetic acid (5% *w/v*) for 5 min. After 10 min of centrifuging the homogenate at 12,000 rpm, 4 mL of the supernatant was collected and combined with 4 mL of aqueous TBA solution 0.02 M. The mixture was incubated at 100 °C in a water bath for 60 min. Further, after the samples reached the room temperature, the absorbance was measured at a wavelength of 532 nm using the Specord 205, Analytik Jena Inc. (Jena, Germany) spectrophotometer. Spectrophotometric measurements were done against a control sample that was sausage-free.

The amount of MDA in the samples was calculated used a calibration curve performed by using MDA solutions with different concentrations in the range 10–50 µg/mL. The results were expressed in µM of MDA/g of sample and reported as the mean value of three independent analyses ± standard deviation (SD).

### 2.7. Statistical Data Analysis

All calculations were done in triplicate, and the results were expressed as mean value ± standard deviation (SD). A one-way ANOVA and a two-sample t-test with equal variances were used to assess the statistical significance of the differences registered between samples. After performing statistical data processing, the results reported within the same row or column in the case of tables, or bars in the case of charts, having different superscripts or

letters are significantly different ($p < 0.05$), while the data that are presented on the same row, column or bars, having the same superscripts or letters, are not significantly different ($p > 0.05$). Microsoft Excel 365 was used to process the data statistically.

## 3. Results and Discussion

### 3.1. Phytochemical Profile of Tomatoes and Tomatoes Processing Byproducts

3.1.1. Assessment of Total Phenolic Content (TPC)

The values for the total polyphenol content are shown in Figure 2. For the total polyphenol content of fresh tomatoes, a content of 11.06 mg GAE/g d.s for LT and 12.48 mg GAE/g d.s for CT were recorded.

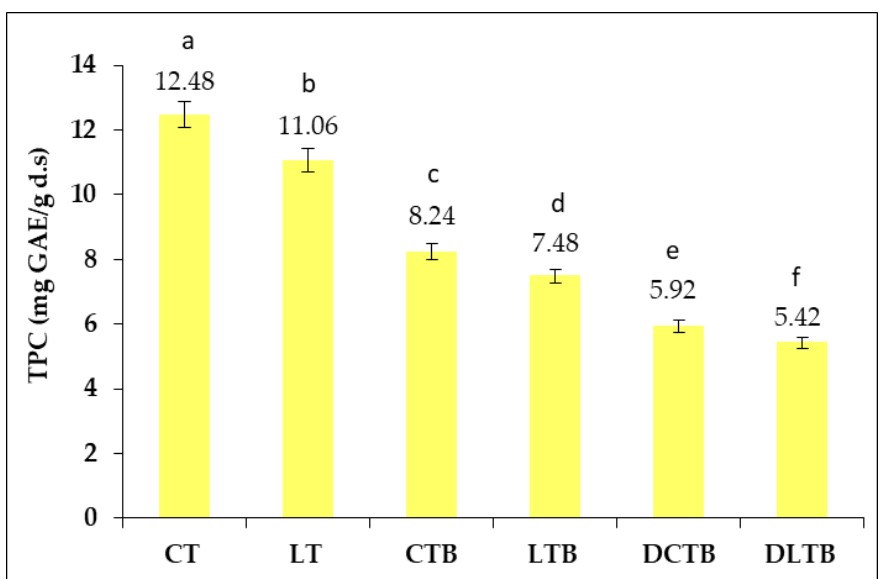

**Figure 2.** Total phenolic content (TPC) of tomatoes and their raw and dried processing byproducts. Results are expressed as the mean value of three independent analyses ± standard deviation (SD) indicated by the error bars. According to the *t*-test, the values for bars sharing different letters (a–f) are significantly different ($p < 0.05$).

As regards the tomato processing byproducts shown in Figure 2, it can be seen that the resulted wastes still maintain a significant level of phenolic compounds (8.24 mg GAE/g d.s for CTB and 7.48 mg GAE/g d.s for LTB), with values 34% and 32% lower than the corresponding fresh tomatoes. Significant values were also recorded for DCTB and DLTB (5.92 mg GAE/g d.s and 5.42 mg GAE/g d.s, respectively) in this case, the values are 28% and 27% lower, respectively, compared to the values recorded for CTB and LTB. Analyzing the obtained values, we can conclude that the valorization of the tomatoes byproducts is viable, considering that they contain significant amounts of total polyphenols. Between the two types of tomatoes, CTB recorded higher values for both the initial product and the byproduct, but also the dried byproduct, compared to LTB. The losses of polyphenols are higher in the case of CTB compared to LTB, both at the separation stage of the byproduct and the stage of dehydration. There were higher CPT losses in technological processes compared to the drying process.

The results are in line with other studies, which show that losses in the total polyphenolic content in response to air drying of tomatoes at 60 °C were in the range of 27% to 33% of the initial values [12]. Regarding the amount of TPC in tomatoes, for cherry tomatoes, other studies reported a content of 41.692 mg GAE/100 g in fresh fruits [31], and for large tomatoes, they reported a content of 259.16–498.60 mg GAE/kg. Silva-Beltrán et al. [32] reported a content of total polyphenols in tomato byproducts between 44.18 and 20.94 mg GAE/100 g, and Peschel et al. [33] reported a content of 61 mg GAE/g dry extract in tomato

peels. Lower TPC content in byproducts compared to whole fruits was also reported by Izzo et al. [31] who observed a content of 41.692 mg GAE/100 g in fresh fruit and 31.750 mg GAE/100 g in tomato skin. The byproducts resulting from the processing of vegetables have high polyphenolic compounds, which is in highly correlation with antioxidant activity [34], making it possible to use them later as food ingredients with a potentially high added value. These data make it possible to improve the knowledge of the stability of polyphenolic compounds during the conditioning of vegetable processing byproducts by drying to preserve their content in bioactive compounds.

### 3.1.2. Chromatographic Evaluation of Individual Polyphenolic Compounds by LC-MS

The content of polyphenolic compounds identified in tomato samples and their raw and dried processing byproducts is given in Table 3.

**Table 3.** Polyphenolic compounds content of tomatoes and tomato processing byproducts.

| Polyphenolic Compound | RT (Min) | Compound Content (mg/g d.s) | | | | | |
|---|---|---|---|---|---|---|---|
| | | CT | LT | CTB | LTB | DCTB | DLTB |
| Gallic acid | 5.694 | 3.481 ± 0.132 [a] | 3.074 ± 0.124 [b] | 3.350 ± 0.126 [a] | 2.940 ± 0.118 [b] | 2.056 ± 0.094 [c] | 1.960 ± 0.091 [c] |
| Protocatechuic acid | 12.631 | 3.579 ± 0.141 [d] | 6.542 ± 0.255 [a] | 3.132 ± 0.152 [e] | 5.234 ± 0.201 [b] | 2.666 ± 0.125 [f] | 4.426 ± 0.191 [c] |
| Caffeic acid | 18.747 | 3.450 ± 0.140 [a] | 2.584 ± 0.122 [c] | 3.336 ± 0.139 [a] | 2.013 ± 0.092 [d] | 3.117 ± 0.118 [b] | 1.973 ± 0.095 [d] |
| Epicatechin | 23.417 | 2.332 ± 0.124 [b] | 2.576 ± 0.126 [a] | 2.213 ± 0.121 [c] | 2.235 ± 0.125 [c] | 1.805 ± 0.920 [e] | 1.951 ± 0.954 [d] |
| *p*-Coumaric acid | 24.952 | 0.376 ± 0.012 [a] | 0.260 ± 0.011 [b] | 0.188 ± 0.010 [c] | 0.133 ± 0.009 [c] | 0.074 ± 0.009 [d] | 0.065 ± 0.008 [d] |
| Ferulic acid | 23.521 | 4.272 ± 0.022 [a] | 2.544 ± 0.015 [d] | 4.011 ± 0.021 [b] | 2.052 ± 0.014 [e] | 3.152 ± 0.012 [c] | 1.252 ± 0.010 [f] |
| Rutin | 25.837 | 24.105 ± 1.075 [a] | 9.311 ± 0.335 [d] | 22.280 ± 1.005 [b] | 8.210 ± 0.257 [e] | 14.852 ± 0.662 [c] | 7.162 ± 0.324 [f] |
| Rosmarinic acid | 28.631 | 1.813 ± 0.090 [a] | 0.615 ± 0.028 [d] | 1.638 ± 0.062 [b] | 0.419 ± 0.020 [e] | 1.282 ± 0.051 [c] | 0.218 ± 0.088 [f] |
| Resveratrol | 29.200 | 3.091 ± 0.138 [a] | 1.639 ± 0.071 [c] | 2.587 ± 0.132 [b] | 1.400 ± 0.603 [d] | 1.598 ± 0623 [c] | 1.351 ± 0.516 [d] |
| Quercetin | 31.871 | 2.361 ± 0.128 [a] | 1.726 ± 0.746 [c] | 2.096 ± 0.904 [b] | 1.563 ± 0.615 [d] | 1.548 ± 0.557 [d] | 1.302 ± 0.413 [e] |
| Kaempferol | 34.644 | 0.750 ± 0.025 [c] | 0.998 ± 0.032 [a] | 0.631 ± 0.022 [d] | 0.892 ± 0.030 [b] | 0.439 ± 0.012 [e] | 0.414 ± 0.010 [e] |

Results are expressed as the mean value of three independent analyses ± standard deviation (SD). The mean differences between the tomatoes and tomato processing byproducts samples were compared using a *t*-test; data from the same row sharing different superscripts (a–f) are significantly different ($p < 0.05$); data from the same row sharing the same superscripts are not significantly different ($p > 0.05$).

Eleven polyphenolic compounds were identified in the analyzed samples. It can be observed that the following polyphenols were identified in the investigated samples, as follows: gallic acid (1.960–3.481 mg/g), protocatechuic acid (2.666–6.542 mg/g), caffeic acid (1.973–3.450 mg/g), epicatechin 1.805–2.576 mg/g), *p*-coumaric acid (0.065–0.376 mg/g), ferulic acid (1.252–4.272 mg/g), rutin (7.162–24.105 mg/g), rosmarinic acid (0.218–1.813 mg/g), resveratrol (1.351–3.091 mg/g), quercitin (1.302–2.361 mg/g) and kaempferol (0.414–0.998 mg/g). It should be noted that for all samples examined, the rutin was found in the largest amount of the detected polyphenolic compounds. Significant amounts were registered for gallic acid, protocatechuic acid, caffeic acid, ferulic acid, epicatechin, resveratrol and quercetin, while kaempferol, rosmarinic acid and *p*-coumaric acid were identified at lower amounts. Except for protocatechuic acid and kaempherol, all other compounds were identified in higher proportions in CT compared to LT. During processing and dehydration, there were registered losses of each polyphenolic compound, and the level of losses was higher during the dehydration process.

For all identified polyphenols, significant differences ($p < 0.05$) were found among samples belonging to the three categories: fresh tomatoes (CT and LT), raw tomato processing byproducts (CTB and LTB) and dehydrated byproducts (DCTB and DLTB).

Silva-Beltrán et al. [32] studied the phenolic compounds from tomato byproducts, identifying gallic acid with the highest values (104.6–463.41 mg/100 g d.s.), followed by chlorogenic acid, rutin, caffeic acid, quercetin and ferulic acid, with levels of only 0.0195–8.09 mg/100 g d.s.).

According to other investigations, Izzo et al. [31] reported that rutin is the most abundant polyphenol in tomatoes, recording a content between 1.191 and 9.516 mg/100 g d.s. In the same study, similar amounts were identified for caffeic acid (0.189–0.894 mg/100 g), *p*-coumaric acid (0.036–0.080 mg/100 g) and ferulic acid (0.256–0.470 mg/100 g), while protocatechuic acid (0.001–0.003 mg/100 g), quercetin (0.009–0.018 mg/100 g) and și kaempferol

(0.003−0 0.007 mg/100 g) were recorded much lower levels. In another study, Perea-Domínguez et al. [35] reported similar values for the identified polyphenolic compounds: caffeic acid (0.25–22.34 mg/g), ferulic acid (1.08–2.53 mg/g), gallic acid (1.25–2.32 mg/g), *p*-coumaric acid (0.15–5.86 mg/g), kaempferol (0.04–0.31 mg/g) and quercetin (0.1–0.85 mg/g), except for the rutin that was found in a much lower quantity (0.08–0.55 mg/g).

### 3.1.3. Lycopene Content

Lycopene is a carotenoid found naturally in many fruits and vegetables. Lycopene is a powerful antioxidant abundant in the blood, liver, skin, prostate, colon and lungs of humans. This substance's antiproliferative and antioxidant activities have been demonstrated in prior studies. According to other research [36,37], eating many lycopene-rich foods lowers the risk of cancer, heart disease and other diseases.

Figure 3 shows the lycopene content of fresh tomato samples and their raw and dried processing byproducts.

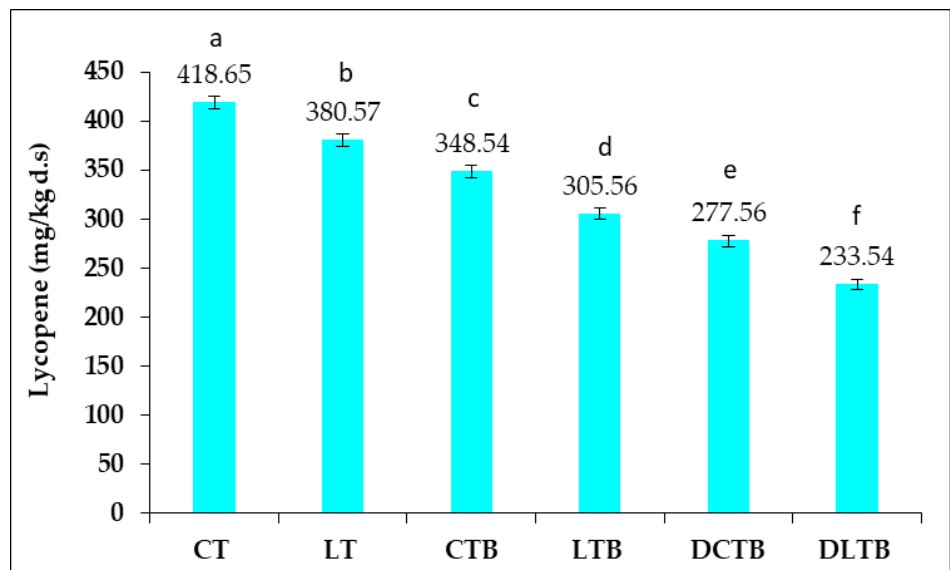

**Figure 3.** Lycopene content of tomatoes and their raw and dried processing byproducts. Results are expressed as the mean value of three independent analyses ± standard deviation (SD) indicated by the error bars. According to the t-test, the values for bars that have different letters (a–f) are significantly different ($p < 0.05$).

The obtained results show that lycopene levels are high in the fresh tomato samples CT (418.65 mg/kg d.s) and LT (380 mg/kg d.s). Additionally, their processing byproducts, CTB and LTB, exhibit considerable amounts of lycopene (348.54 and 305.56 mg/kg d.s, respectively), with values of 16% and 19% lower than those of fresh tomatoes.

Our results consistent with those of other researchers, who found in tomatoes a lycopene content of 38 mg/100 g d.s [38]. In fresh tomatoes Martínez-Valverde er al. [39], a lycopene content of between 18.60–64.98 mg/kg was reported. In fresh tomatoes Martínez-Valverde er al. [39], reported a lycopene content between 18.60–64.98 mg/kg.

The lycopene content of raw tomato processing byproducts CTB and LTB was reduced by 20% and 23%, respectively, due to convective drying compared to the initial values. Thus, to maintain high lycopene content, our results indicate that a moderate temperature of 60 °C is recommended for drying tomato processing byproducts. The only drawback of drying at 60 °C is the long drying time of tomato byproducts.

### 3.2. The Proximate Composition of Sausages

The results presented in Table 4 illustrate the proximate composition of sausages.

**Table 4.** Proximate composition of sausage formulas.

| Sample | Chemical Parameters | | | | | | |
|---|---|---|---|---|---|---|---|
| | Moisture (g/100 g) | Protein (g/100 g) | Lipids (g/100 g) | Ash (g/100 g) | NaCl (g/100 g) | Carbohy-Drates (g/100 g) | Energy Value (kcal/100 g) |
| SC | 49.583 ± 1.354 [a] | 14.670 ± 0.361 [l] | 30.662 ± 0.761 [d] | 2.290 ± 0.051 [h,i] | 2.110 ± 0.055 [a] | 0.685 | 337.378 |
| SDC | 48.982 ± 1.300 [b,d] | 15.430 ± 0.379 [k] | 30.540 ± 0.768 [a] | 2.380 ± 0.050 [h,i] | 2.120 ± 0.058, [a] | 0.548 | 338.772 |
| SDCN | 48.974 ± 1.299 [b,d] | 15.410 ± 0.352 [k] | 30.482 ± 0.755 [a,b] | 2.360 ± 0.048 [i] | 2.130 ± 0.056 [a] | 0.644 | 338.554 |
| SSC | 48.758 ± 1.315 [b,d,e] | 15.350 ± 0.344 [k] | 31.031 ± 0.723 [c] | 2.450 ± 0.055 [f,g] | 2.120 ± 0.055 [a] | 0.291 | 341.843 |
| SSCN | 48.687 ± 1.218 [b,d,f] | 15.400 ± 0.358 [k] | 31.102 ± 0.729 [b,c] | 2.410 ± 0.050 [g,h] | 2.140 ± 0.056 [a] | 0.261 | 342.562 |
| DSDLTB50 | 48.933 ± 1.323 [b] | 16.180 ± 0.361 [j] | 29.242 ± 0.705 [e] | 2.570 ± 0.058 [e,f] | 2.130 ± 0.065 [a] | 0.945 | 331.677 |
| DSDLTB90 | 48.837 ± 1.315 [b,d] | 16.540 ± 0.375 [i] | 28.971 ± 0.695 [e,f] | 2.660 ± 0.060 [d] | 2.110 ± 0.063 [a] | 0.881 | 330.427 |
| DSDLTB180 | 48.573 ± 1.309 [b,d,f] | 17.460 ± 0.385 [f,g] | 28.349 ± 0.694 [g,h] | 2.700 ± 0.065 [c] | 2.140 ± 0.055 [a] | 0.778 | 328.093 |
| DSDLTB270 | 48.312 ± 1.297 [e,f] | 18.110 ± 0.388 [c] | 27.974 ± 0.681 [h] | 2.890 ± 0.074 [b] | 2.120 ± 0.054 [a] | 0.594 | 326.580 |
| SSDLTB50 | 48.831 ± 1.285 [b,d] | 16.460 ± 0.399 [i] | 29.213 ± 0.709 [e] | 2.480 ± 0.056 [e,f] | 2.140 ± 0.052 [a] | 0.876 | 332.261 |
| SSDLTB90 | 48.563 ± 1.313 [b,d,f] | 16.980 ± 0.325 [h] | 28.748 ± 0.676 [f] | 2.690 ± 0.059 [d] | 2.130 ± 0.055 [a] | 0.889 | 330.208 |
| SSDLTB180 | 48.210 ± 1.334 [c,f] | 17.680 ± 0.314 [d] | 28.342 ± 0.655 [g,h] | 2.840 ± 0.066 [c] | 2.120 ± 0.058 [a] | 0.808 | 329.031 |
| SSDLTB270 | 47.947 ± 1.275 [c] | 18.070 ± 0.379 [c] | 27.952 ± 0.611 [h] | 2.910 ± 0.075 [a,b] | 2.140 ± 0.052 [a] | 0.981 | 327.772 |
| DSDCTB50 | 48.746 ± 1.319 [b,d] | 17.080 ± 0.305 [g,h] | 28.941 ± 0.714 [e,f] | 2.410 ± 0.060 [e] | 2.110 ± 0.050 [a] | 0.713 | 331.641 |
| DSDCTB90 | 48.689 ± 1.307 [b,d,f] | 17.370 ± 0.311 [e,f] | 28.754 ±0.707 [f] | 2.680 ± 0.061 [d] | 2.140 ± 0.057 [a] | 0.367 | 329.734 |
| DSDCTB180 | 48.358 ± 1.296 [c,d,e] | 17.960 ± 0.365 [c] | 28.069 ± 0.653 [h] | 2.750 ± 0.066 [c] | 2.130 ± 0.052 [a] | 0.733 | 327.395 |
| DSDCTB270 | 48.024 ± 1.285 [c] | 18.930 ± 0.391 [a] | 27.164 ± 0.587 [i] | 2.920 ± 0.077 [a,b] | 2.120 ± 0.054 [a] | 0.842 | 323.564 |
| SSDCTB50 | 48.970 ± 1.324 [b,d] | 16.890 ± 0.358 [h] | 28.856 ± 0.713 [e,f] | 2.560 ± 0.057 [e,f] | 2.110 ± 0.055 [a] | 0.614 | 329.720 |
| SSDCTB90 | 48.537 ± 1.316 [b,d,f] | 17.480 ± 0.322 [d,e] | 28.564 ± 0.706 [f,g] | 2.670 ± 0.060 [d] | 2.140 ± 0.053 [a] | 0.609 | 329.430 |
| SSDCTB180 | 48.126 ± 1.303 [c,f] | 17.990 ± 0.368 [c] | 28.221 ± 0.641 [g,h] | 2.820 ± 0.068 [c] | 2.120 ± 0.055 [a] | 0.723 | 328.841 |
| SSDCTB270 | 47.958 ± 1.288 [c] | 18.440 ± 0.381 [b] | 27.996 ± 0.628 [h] | 2.950 ± 0.080 [a] | 2.130 ± 0.058 [a] | 0.526 | 327.828 |

Results are expressed as the mean value of three independent analyses ± standard deviation (SD). The mean differences between the sausage formulas (both control samples and sausages samples supplemented with different doses of dried tomatoes byproducts), were compared using a *t*-test; data from the same column sharing different superscripts (a–l) are significantly different ($p < 0.05$); data from the same column sharing the same superscripts are not significantly different ($p > 0.05$).

Table 2 shows the proximate composition (moisture, crude protein, fat and total ash) of the studied sausage samples. The moisture content of the analyzed samples ($p < 0.05$) varied between 47.947 g/100 g and 49.583 g/100 g, being significantly influenced by the treatments applied and the inclusion of tomato byproducts DLTB and DCTB. The moisture content was significantly higher in the control sample (C: 49.583 g/100 g) compared to the control sausage samples thermally treated by smoking and drying (SDC: 48.982 g/100 g and SDCN: 48.974 g/100 g), but also the baked and smoking samples (SSC: 48.758 g/100 g and SNA: 48.687 g/100 g). Additionally, significant differences ($p < 0.05$) in terms of humidity content were also recorded between the control sample (C: 49.583 g/100 g) and the samples of sausages enriched with DLTB (DSDLTB 50, 90, 180 and 270: 48.933–48.312 g/100 g; SSDLTB 50, 90, 180 and 270: 48.831–47.947 g/100 g) and DCTB (DSDCTB50, 90, 180 and 270: 48.746–48.024 g/100 g; SSDLTB 50, 90, 180 and 270: 48.970–47.958 g/100 g). This finding confirms previous studies by Ba et al. [40], who reported that the moisture content of sausages is approximately 42.91–45.91%. The lower moisture content observed in samples fortified with DLTB and DCTB was most likely due to the inclusion of these dehydrated byproducts in sausage formulae and byproducts with low moisture content. Similarly, Balzan et al. [41] reported a moisture content for fresh sausages between 38.0 and 54.4%, which is inversely proportional to the shelf life of the products. Additionally, in the same study, a lower humidity level is reported (37.7–43.8%) for the samples subjected to the cooking treatment.

Due to a close association between water molecules and the hemoproteins in meat, protein is essential for keeping water in the meat [42]. Tomato byproducts, consisting of skins and seeds, after drying and grinding, are characterized by a significant intake of

protein (175.6 g/kg) [43], which influences the composition of sausages with the addition of tomato byproducts. A slight gradual increase in the protein content in sausages was observed as the level of addition of DLTB and DCTB increased. Thus, the protein levels in the studied samples were between 14.670 g/100 g and 18,930 g/100 g, with significant differences ($p < 0.05$) between the blank samples (C, DCN, DC, SCN, SC, SC and SC with values within the limits of 14.670–15.430 g/100 g) and the samples with the addition of tomato byproducts (DSDLTB, SSDLTB, DSDCTB and DSDCTB with values within the limits of 16.180–18.930 g/100 g). Similar trends were observed by Fernández-Ginés et al. [44], who reported a higher protein content in salami with added citrus fiber than the control sample. In contrast, Sam et al. [42] reported a decrease in the protein content with the addition of carrots in frankfurter-type sausage.

The lipid content in the dried and ground tomatoes byproduct is 95.9 g/kg [43]. Thus, the inclusion of these byproducts (DLTB and DCTB) in different doses in the sausage formulas significantly influenced ($p < 0.05$) the fat content, with the control samples having the highest percentages of fat (30.662 g/100 g). A slight reduction in fat content was observed as the level of addition of DLTB and DCTB increased (Table 2). Of the samples fortified with DLTB, SSDLTB270 had the lowest fat content (47.864 g/100 g), and of the samples with the addition of DCTB, SSDCTB270 had the lowest fat level (27.696 g/100 g).

Concerning the heat treatments applied, it can be noted that the type of treatment applied thermally does not significantly influence the fat content. Sam et al. [42] reported similar findings when different doses of carrot paste were added in the frankfurter sausages recipe. Wang et al. [45] reported a reduction in the fat level for sausages in which they added Tomato peel powder compared to the control sample without adding Tomato peel powder. The reduction of the content of fat enriched with CP is a positive result since they are more likely to suffer the oxidation of lipids very slowly.

The ash content is used as a measure of the total mineral content [42]. The content of mineral substances (Table 2) was within the limits of 2.360–2.950 g/100 g, being significantly higher ($p < 0.05$) in the case of sausage samples with the addition of tomato byproducts, namely DSDLTB, SSDLTB, DSDCTB and SSDCTB. It can also be seen that the content of mineral substances increases proportionally with the dose of DLTB and DCTB added in the sausage formulas.

Similar results were also reported by Ahmad et al. [46], who added various vegetables (oyster mushrooms, purple cabbage, spinach, bell peppers and carrot) to chicken sausages and attributed the results to the high fiber content in vegetables. Sam et al. [42] reported a similar trend with adding carrot paste to sausages.

The NaCl content was within the limits of 2.10–2.14 g/100 g; for this indicator, there were no significant differences between the studied sausage samples. The quantities of NaCl recorded for the investigated samples are consistent with the values reported in the literature; namely, Ellekjær et al. [47] reported for sausages a NaCl content between 1.4 and 2.2%, and Aaslyng et al. [48] reported a NaCl content between 1.74 and 2.19%.

The carbohydrate content recorded values between 0.261 and 0.945 g/100 g, varying without being influenced by the heat treatment applied or the addition of byproducts in the composition of the sausages. Romero et al. [49] reported similar carbohydrate values for different types of sausages, respectively, 0.63–2.33%.

Another advantage of adding DLTB and DCTB to sausages is the reduction of the energy value, the sausage samples with the addition of tomato byproducts had a lower energy value (326.58–332.26 kcal/100 g) than the control samples that were within the limits of 337.38–342.56 kcal/100 g. These values fall within the limits reported by Romero et al. [49] for different types of sausages (261.14–376.27 kcal/100 g).

### 3.3. Oxidative Stability Assessment

Oxidation of lipids is the main factor of damage to meat products both during processing and storage. This process leads to the formation of dangerous chemicals such as 4-hydroxytoluene as well as the generation of unpleasant smells, rancid odors, color

changes and other side effects [50]. Oxidative stability of manufactured sausage formulas was investigated during the storage period by assessing the degree of primary and secondary lipid oxidation. Thus, the peroxide value (PV) was used as a measure of primary lipid oxidation, while the secondary lipid oxidation was revealed by determining p-anisidine value (*p*-AV) and thiobarbituric acid (TBA) value.

### 3.3.1. Peroxide Value (PV)

One of the most frequent tests for determining primary oxidation in food is the peroxide value. This study determined the degree of oxidation by measuring the peroxide value of sausage samples kept at 4 °C for 20 days. Figure 4 shows the effect of sausage formulas supplementation with dried tomato processing byproducts (DLTB and DCTB) on the peroxide value during storage, compared to the control sample and the sausage samples to which sodium nitrite was added.

During the lipid oxidation process, unsaturated fatty acids oxidize, leading to the formation of hydroperoxides that quickly decompose into secondary compounds, including hydrocarbons, ketones, aldehydes, alcohols, acids and esters, which are compounds that cause the formation of unpleasant odor in meat products [51].

PV was significantly higher in the case of control samples (SC, SDC and SSC) compared to samples with the addition of sodium nitrite, DLTB and DCTB, throughout the 20 days of storage.

During the storage, the values of the peroxide index were influenced both by the addition of DLTB and DCTB and by the thermal treatments applied to the sausages (Figure 4). Thus, PV decreased with the increase in the dose of DLTB and DCTB used in the sausage formulas. Additionally, lower PV values were recorded for sausage samples with the addition of DCTB (0.392–4.048 meq $O_2$/kg) compared to those with added DLTB (0.300–3.559 meq $O_2$/kg). This is due to the higher content of antioxidant compounds (polyphenols and lycopene) of DCTB compared to DLTB.

Regarding the influence of heat treatment, it can be noted that the samples that were subjected to the scalding process recorded lower values (0.300–3.328 meq $O_2$/kg for SSDCTB and 0.392–3.867 meq $O_2$/kg for SSDLTB) compared to those that were dried (0.469–3.559 meq $O_2$/kg for DSDCTB and 0.485–4.048 meq $O_2$/kg). Significant differences ($p < 0.05$) between PV of investigated sausage samples were recorded for all storage periods. Comparing the PV recorded for the sodium nitrite control samples (DCN and SNA) and the samples with the addition of DLTB and DCTB, we can conclude that DCTB at doses providing a polyphenolic compounds level of 180 mg GAE/kg of raw processed meat and 270 mg GAE/kg of processed meat, respectively, can replace the sodium nitrite in meat products for both dry and cooked sausage formulas. Encouraging results were also recorded when DLTB was incorporated in sausage formulas at a dose that ensures a content of polyphenolic compounds of 270 mg GAE/kg of raw processed meat.

The results of our study were consistent with those of Jouki et al. [52], who reported that tomato paste reduced PV in chicken sausages when different concentrations of tomato paste (1, 2 and 10%) were added compared to the blank sample, during storage at 4 °C for 14 days. They reported 2.83 meq $O_2$/kg for the control sample on day 1 and 15.17 meq $O_2$/kg on day 14, and for samples with the addition of tomatoes between 2.19 meq $O_2$/kg on day 1 and 7.83 meq $O_2$/kg on day 14. Moreover, in this case, the proportion of tomato paste incorporated in the chicken sausages directly influenced the PV.

Additionally, Sam et al. [42] reduced PV over 14 days of storage in the case of frankfurter sausages by adding carrot paste in different proportions compared to the control sample.

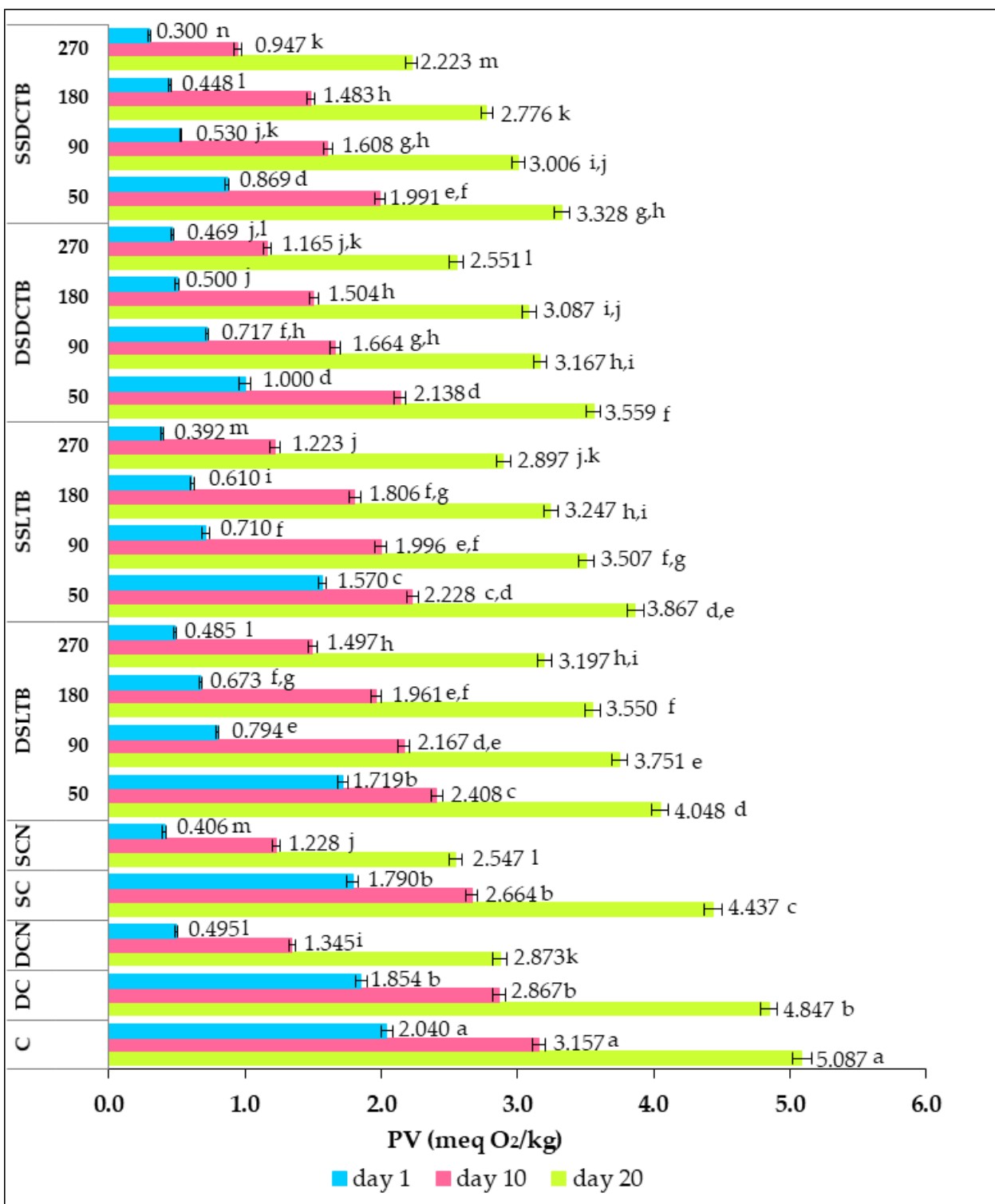

**Figure 4.** The effect of sausage formulas supplementation with dried tomato processing byproducts on peroxide value (PV) during storage. Results are expressed as the mean value of three independent analyses ± standard deviation (SD) indicated by the error bars. The mean differences between the sausage formulas (both control and sausages samples supplemented with dried tomatoes byproducts), from each storage period (day 1, 10 and 20) were compared using a t-test; data from the same day sharing different superscripts (a–n) are significantly different ($p < 0.05$); data from the same day sharing the same superscripts are not significantly different ($p > 0.05$).

### 3.3.2. *p*-Anisidine Value (*p*-AV)

Lipid hydroperoxides can break down to form MDA and a range of aldehydes such as alkanals, 2-alkenals and 2,4-alkadienals. In secondary lipid oxidation, *p*-anisidine is a frequently used spectroscopic method for measuring carbonyls, especially unsaturated aldehydes. The reversible aldehyde carbonyl bond of the *p*-anisidine amine group is responsible for catalyzing the reaction. The reaction's byproduct is a Schiff base with a maximum absorbance of 350 nm. The *p*-anisidine value is a good indicator of the oxidation of meat and meat products since it correlates well with other indicators of primary oxidation (peroxide value) and secondary oxidation (TBAR and volatile aldehydes), as well as with the degradation of organoleptic qualities. The dissolution of lipid hydroperoxides is excluded [51]. In addition to MDA, lipid hydroperoxides decompose to produce a variety of aldehydes, such as alkanals, 2-alkenals and 2,4-alkadienals. *p*-anisidine is a spectroscopic technique, frequently used to quantify carbonyls, particularly unsaturated aldehydes, in secondary lipid oxidation. The *p*-anisidine amine group's aldehyde carbonyl bond, which is reactive, is the catalyst for the reaction. The Schiff base generated in this reaction shows an absorbance maximum at 350 nm [52].

Figure 5 shows the effect of sausage formulas supplementation with dried tomato processing byproducts on *p*-anisidine value (*p*-AV) during storage for 1, 10 and 20 days.

A closer look on the obtained results reveals that the control samples (SC, SDC and SSC) recorded significant *p*-AV ($p < 0.05$) higher (1.7–6–4.268) compared to the sausage samples with the addition of sodium nitrite (0.3–8–1.587), respectively, compared to those supplemented with DLTB (0.2–5–2.443) and DCTB (0.2–2–2.150) during 20 days of storage.

The *p*-AV was influenced during storage by adding DLTB and DCTB and the heat treatment applied to the sausages (Figure 4). Thus, *p*-AV decreased with the increase in the dose of DLTB and DCTB incorporated in the sausage formulas. Additionally, lower *p*-AV were recorded for sausage samples with the addition of DCTB (0.202–2.150) compared to those with the addition of DLTB (0.275–2.443) due to the high content of antioxidant compounds (polyphenols and lycopene) found in DCTB compared to DLTB. Regarding the influence of heat treatment, it can be noted that the samples that were subjected to the scalded process (SSDLTB and SSDCTB) recorded lower values (0.202–2.241) compared to those that were dried (DSDLTB and DSDCTB) (0.260–2.443). These significant differences ($p < 0.05$) between the samples of the sausages analyzed were recorded during the entire storage period. Comparing the *p*-AV recorded for the control samples with addition of sodium nitrite (DCN and SNA) and the samples with DLTB and DCTB, we can conclude that DCTB at doses that provide a polyphenolic compounds content of 180 mg GAE/kg of processed meat and 8270 mg GAE/kg of processed meat, respectively, can replace the sodium nitrite in meat products for both dry and boiled samples. Promising results were also recorded by incorporation of DLTB into sausage formulas at a dose that ensures a polyphenolic compounds level of 270 mg GAE/kg of processed meat.

A similar trend in applying different heat treatments to meat products was also reported by Rasinska et al. [53], who analyzed the degree of lipid oxidation of meat for which different heat treatments were applied. Thus, in this case, for the meat to which the scalded treatment was applied, lower *p*-AV were obtained compared to the dried samples. Cocan et al. [29], reported lower values for *p*-AV in oil samples supplemented with sweet pepper seed oil and chili oil compared to samples without addition.

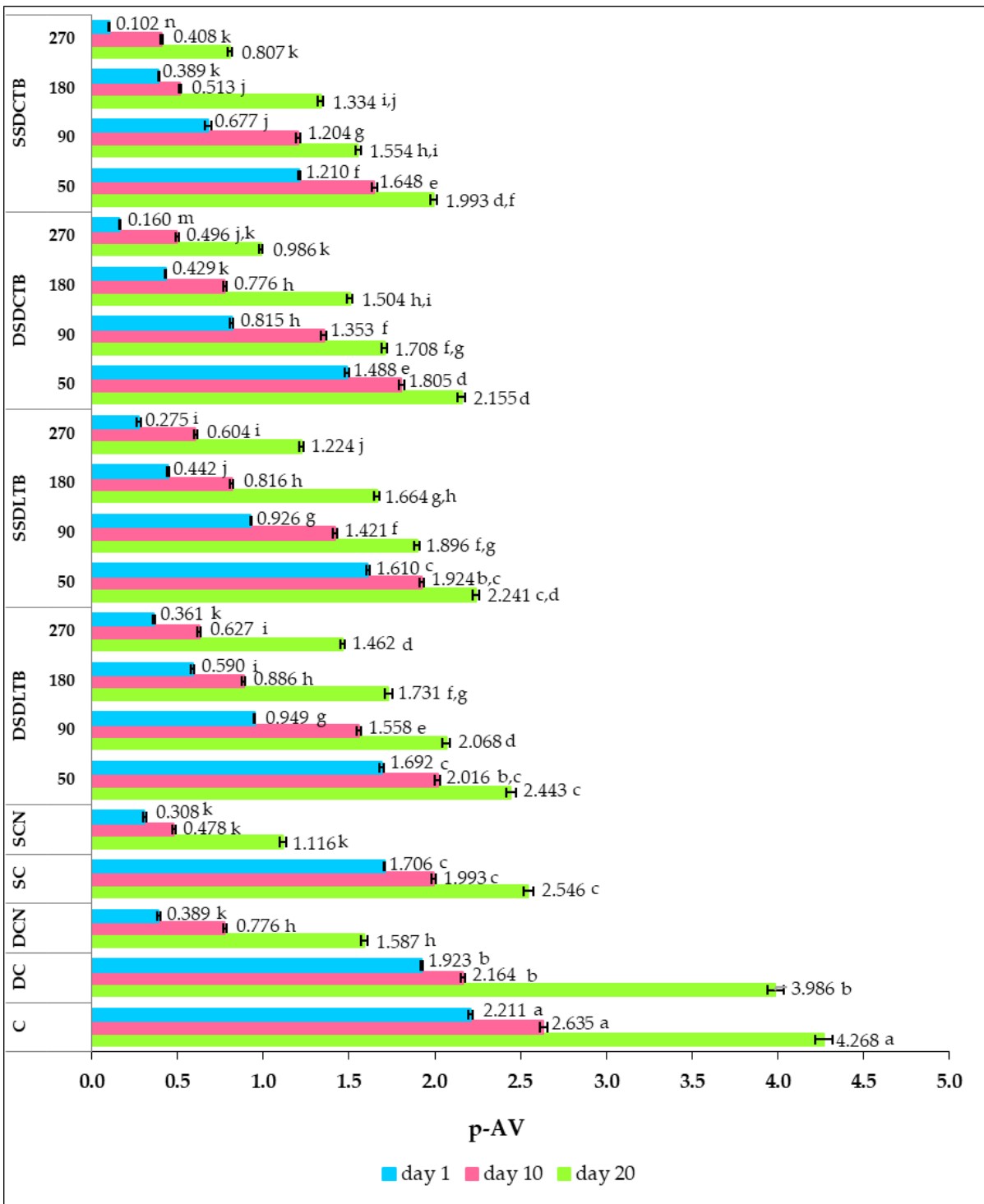

**Figure 5.** The effect of sausage formulas supplementation with dried tomato processing byproducts on *p*-anisidine value (*p*-AV) during storage. Results are expressed as the mean value of three independent analyses ± SD indicated by the error bars. The mean differences between the sausage formulas (both control and sausages samples supplemented with dried tomatoes byproducts), from each storage period (day 1, 10 and 20) were compared using a *t*-test; data from the same day sharing different superscripts letters are significantly different ($p < 0.05$); data from the same day sharing the same superscripts are not significantly different ($p > 0.05$).

### 3.3.3. Total Oxidation Value (TOTOX)

In general, it is not advisable to examine a single oxidation indicator because this does not provide accurate and comprehensive information about the oxidation state of a sample. Combining primary and secondary lipid oxidation assays is more beneficial since they provide information on the current oxidative status and the full oxidation process (primary and secondary). Considering this, even though this is not an analytical method in and of itself, it was advised to compute the total oxidation of the beef samples using the TOTOX value. This index is made up of both PV and *p*-AV, as the sum of twice of PV and *p*-AV. The recommended formula is predicated on the idea that a two-unit rise in *p*-AV is equivalent to a one-unit increase in PV [54].

In Figure 6 are presented the changes induced in TOTOX values as effect of sausage formulas supplementation with dried tomato processing byproducts (DLTB and DCTB), compared to the control sample and the sausage samples to which sodium nitrite was added. For this indicator, the control samples (SC, SDC and SSC) recorded significantly higher values (5.286–14.442) than the samples with addition of sodium nitrite (1.120–7.333) and those supplemented with DLTB (1.059–10.539) and DCTB (0.702–9.273), respectively, during the 20 days of storage.

As can be seen from the data presented in Figure 6, TOTOX values were influenced during the storage both by the dose of DLTB and DCTB incorporated into the sausage formulas and by the thermal treatments applied during the technological process.

It was noted that TOTOX values decreased with increasing dose of DLTB and DCTB. Lower TOTOX values were recorded for the sausage samples with addition of DCTB (0.702–9.273) compared to the samples supplemented with DLTB (1.059–10.539). Additionally, the influence of thermal treatment is well emphasized, resulting in lower TOTOX values for the scalded samples (SSDLTB and SSDCTB) (0.702–9.975) compared to the dried samples (DSDLTB and DSDCTB) (1.098–10.539).

Comparing the TOTOX values recorded for sodium nitrite control samples (DCN and SCN) and samples with the addition of DLTB and DCTB, we can state that the incorporation of DCTB in sausage formulas at doses that ensure a polyphenolic compounds level of 180 mg GAE/kg of processed meat, respectively 270 mg GAE/kg of processed meat can replace sodium nitrite for both dried sausages and scalded ones. Regarding the supplementation of sausage formulas with DLTB it was noted that this tomato processing byproduct can replace sodium nitrite when is used at a dose providing a polyphenolic compounds content of 270 mg GAE/kg of processed meat.

Similar values also reported Nacak et al. [55], in his study in which they follow the evolution of the degree of oxidation of sausages with the addition of rosemary extract and tocopherol for 3 months. After 30 days the values recorded for TOTOX were within the limits of 5.78–8.61 while the control sample recorded a value of 12.07.

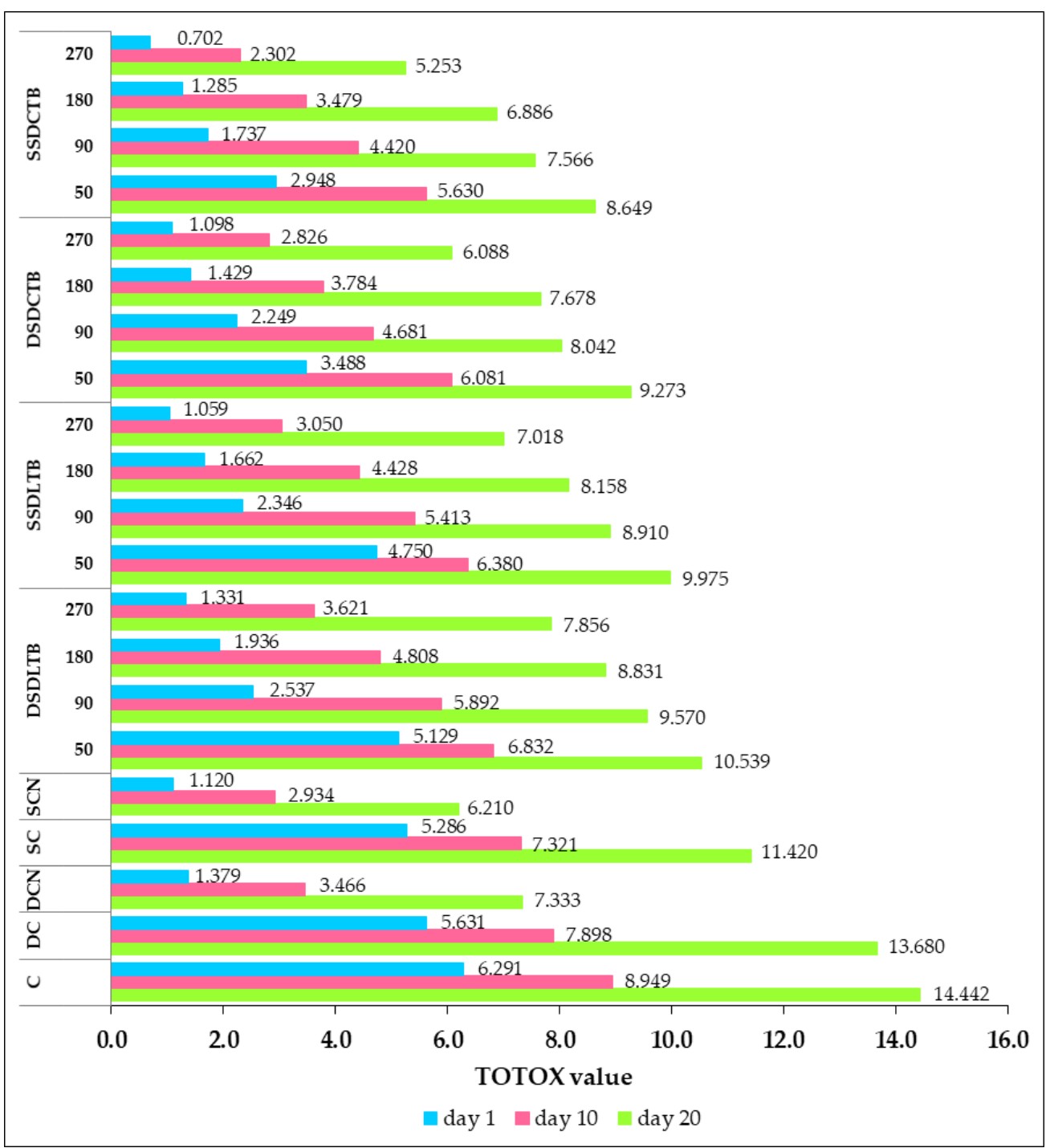

**Figure 6.** The effect of sausage formulas supplementation with dried tomato processing byproducts on TOTOX value during storage.

### 3.3.4. Thiobarbituric Acid (TBA) Value

Malondialdehyde (MDA) is one of the most significant aldehydes formed during the secondary lipid oxidation of polyunsaturated fatty acids (1–3–propanedial). Since it is the principal indicator of lipid oxidation and produces rancid odors in low quantities, this aldehyde is also particularly important in meat. Several studies found that−2–3 mg MDA/kg is the appropriate amount for meat and meat products to be rancidity free [54,56].

Figure 7 shows the effect of sausage formulas supplementation with tomato processing byproducts (DLTB and DCTB) on the TBA value during storage, compared to the control samples and the sausage samples to which sodium nitrite was added.

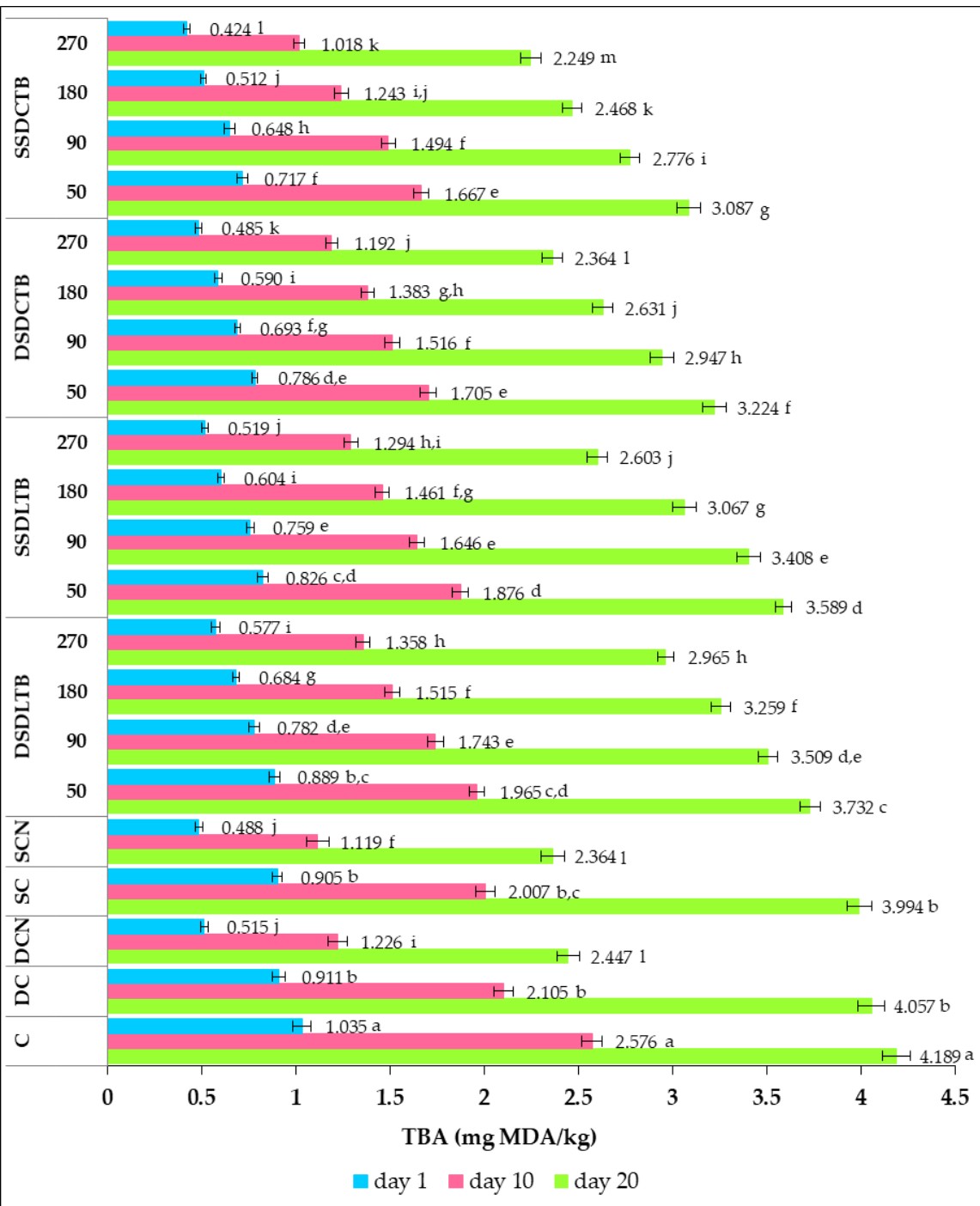

**Figure 7.** The effect of sausage formulas supplementation with dried tomato processing byproducts on thiobarbituric acid (TBA) value during storage. Results are expressed as the mean value of three independent analyses ± standard deviation (SD) indicated by the error bars. The mean differences between the sausage formulas (both control and sausages samples supplemented with dried tomato byproducts), from each storage period (day 1, 10 and 20) were compared using a *t*-test; data from the same day sharing different superscripts (a–m) are significantly different ($p < 0.05$); data from the same day sharing the same superscripts are not significantly different ($p > 0.05$).

TBA values were significantly higher in the case of control samples (SC, SDC and SSC) compared to samples with the addition of sodium nitrite, DLTB and DCTB, throughout the 20 days of storage. TBA values were influenced during storage by adding DLTB and DCTB and the heat treatment applied to sausages (Figure 7).

Thus, TBA values of the investigated sausages samples decreased with the increase of the dose of DLTB and DCTB added in the manufacturing recipe. Additionally, lower values of TBA were recorded for sausage samples with the addition of DCTB (0.424–3.224 mg MDA/kg) compared to those with added DLTB (0.519–3.732 mg MDA/kg). Concerning the influence of heat treatment, in this case, too, it can be noted that the samples that were subjected to the scalded process recorded lower TBA values (0.424–3.087 mg MDA/kg for SSDCTB and 0.519–3.589 mg MDA/kg for SSDLTB) compared to the dried samples (0.485–3.224 mg MDA/kg for DSDCTB and 0.577–3.732 mg MDA/kg). These significant differences ($p < 0.05$) between the analyzed sausage samples were recorded from the first day of storage until day 20.

Comparing the TBA values recorded for the sausages control samples with sodium nitrite (DCN and SCN) and the sausages samples supplemented with DLTB and DCTB, it can be concluded that DCTB at doses providing a level of polyphenolic content of 180, and 270 mg GAE/kg of raw processed meat, respectively, can replace sodium nitrite in meat products for both dry and boiled samples. In a similar manner, to obtain the same effect, DLTB can replace sodium nitrite in the sausage formulas at a dose that ensure a level of polyphenolic compounds of 270 mg GAE/kg of raw processed meat.

For all samples up to 10 days of storage, the MDA values were below 3 mg/kg, suggesting that they were not affected by oxidation and rancidity. After storage for 20 days of scalded sausage samples with addition of DCTB at doses that provide a level of polyphenolic compounds of 180 and 270 mg GAE/kg, respectively, of raw processed meat, and values below 3 mg MDA/kg were recorded, which proves that these samples were protected against oxidation and rancidity. The same effect was found in dried sausage samples supplemented with DCTB at a dose that ensures a level of polyphenolic compounds of 270 mg GAE/kg of processed meat.

In the case of DLTB incorporation in the scalded sausage formulas at doses providing a level of polyphenolic compounds of 180, respectively 270 mg GAE/kg of processed meat, the oxidation and rancidity process was not found to occur. The dried sausage samples were protected against rancidity and oxidation by addition of DLTB at a dose that ensures a level of polyphenolic content of 270 mg GAE/kg of processed meat.

Our results are in agreement with those previously reported [40], when the effect of shiitake extracts byproducts on the oxidation of the sausages during storage for 40 days was investigated. After 20 days of storage for sausages with the addition of shiitake byproducts, the values recorded were between 1.40 and 1.74 mg MDA/kg; for the control sample, they were 2.55 mg MDA/kg; and for the sample with sodium nitrite, they were 1.62 mg MDA/kg.

A similar trend was also recorded by Jouki et al. [52], who studied the effect of the addition of pectin and tomato paste as a natural antioxidant on inhibiting lipid oxidation and producing functional sausages from chicken breast. In this case, the addition of pectin and tomato paste reduced the oxidation degree of the analyzed samples, and the recorded MDA values decreased with the increase in the dose of pectin and tomato paste added in the sausage manufacturing recipe.

## 4. Conclusions

The present study has shown that the recovery of tomato processing byproduct and its subsequent incorposation, in the powder form, in different doses, as natural additive in the recipe for the manufacture of pork meat sausages recipe did not negatively impacted on investigated parameters of the designed products. Sausage formulas supplemented with DLTB and DCTB had a higher protein and ash content and a lower fat content. In addition, the supplementation of sausages with DLTN and DCTB led to products

with higher oxidative stability, from the point of view of both primary and secondary oxidation. In the sausage samples with doses of DCTB providing a level of polyphenolic compounds of 180 mg GAE/kg of processed meat, and 270 mg GAE/kg of processed meat, respectively, the values recorded for PV, *p*-AV, TOTOX value and TBA are comparable to those recorded in sausage samples with the addition of sodium nitrite. In the sausage formulas in which DLTB was incorporated at a dose that ensure a level of polyphenolic compounds of 270 mg GAE/kg of raw processed meat, similar values were recorded to those obtained in the case of sausage samples with the addition of sodium nitrite. In conclusion, tomato processing byproducts could be an attractive natural substitute for sodium nitrite commonly used in the meat products' manufacturing recipes.

Nowadays, when there is an obvious trend to use agro-food processing byproducts as a source of functional components, our study revealed the potential of tomato processing byproducts to be used as a sustainable resource to design innovative meat product formulas by replacing nitrite in the manufacturing recipe of two types of pork sausages: smoked and dried and smoked and scalded, respectively. The reported results highlight the potential of tomato processing byproducts to protect the sausage formulas against lipid oxidation, and our data may also be useful for improving the functional value of meat products. Moreover, this study revealed the possibility to exploit the bioactive potential of tomato processing byproducts as a strategy to reformulate innovative nitrite-free sausages.

**Author Contributions:** Conceptualization, A.I.C., I.C. and M.-A.P.; methodology, A.I.C., I.C., M.N., E.A., D.O., I.H., I.R. and M.-A.P.; formal analysis, A.I.C., I.C., M.N., D.O. and I.H.; writing—original draft preparation, A.I.C., I.C., M.N., E.A., D.O., I.R. and M.-A.P.; writing—review and editing, A.I.C., I.C., M.N., E.A., D.O., I.R. and M.-A.P.; supervision, I.C., E.A., I.R., M.-A.P., A.I.C., I.C., M.N., E.A., D.O.; I.R. and M.-A.P.; funding acquisition, I.R. All authors have read and agreed to the published version of the manuscript.

**Funding:** This research paper is supported by the project "Increasing the impact of excellence research on the capacity for innovation and technology transfer within USAMVB Timisoara", project code 6PFE, submitted in the competition Program 1—Development of the national system of research—development, Subprogram 1.2—Institutional performance, Institutional development projects—Development projects of excellence in R.D.I.

**Institutional Review Board Statement:** Not applicable.

**Informed Consent Statement:** Not applicable.

**Data Availability Statement:** The report of the analyzes performed for the samples in the paper can be found at the Interdisciplinary Research Platform (PCI) belonging to the University of Life Sciences "King Michael I" from Timisoara.

**Acknowledgments:** We have been able to perform this research with the support of the Interdisciplinary Research Platform belonging to the University of Life Sciences "King Michael I" from Timisoara, where the analysis were made.

**Conflicts of Interest:** The authors declare no conflict of interest.

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
