# Peer review of "Exploring the Potential of Tomato Processing Byproduct as a Natural Antioxidant in Reformulated Nitrite-Free Sausages"

_sustainability, doi:10.3390/su141911802_

Round 1
Reviewer 1 Report
The manuscript explores the use of tomato byproducts as antioxidant preservatives in sausages. The topic is well approached and the work was oriented towards a practical application. Moreover, the introduction provides a clear and succinct overview of the state-of-the-art. In regards to the methodology, I have the following questions:
1. Line 94 comments about “sensory analysis” being performed. I kindly disagree with authors, as there was no analysis of this kind in the text. Authors very briefly comment about sensory aspects in the analysis of proximate composition, though it only tackles moisture influence on texture. In this sense, I suggest authors to rephrase the sentences where “sensory analysis” was mentioned;
2. I suggest adding a table to assist readers to clearly differentiate the formula of each sausage preparation;
3. I kindly ask authors to re-check the headings and their numbers. Results were provided with their respective discussions, so no heading 4 was used and there was a number gap between section 3 to 5 (conclusion);
4. Even though Folin-Ciocalteu is extensively described as a test for phenolics, it is also recognized to measure the reducing power of the sample, what has many implications in antioxidant activity. Authors could explore it a bit further on sections 3.1.1. and 3.1.2., as well as better discuss the variations in the phenolic contents under the light of the processes that each test sample underwent;
5. Owing to the amount of data, did authors consider using principal component analysis (PCA) to reduce dimensions and draw information about the dataset in a more succinct manner?
Reviewer 2 Report
Nice study.
Clearly elaborated.
Need Minor corrections as mentioned in the attached file.
Can you firmly suggest that natural products from the crops like tomato can be a better alternative of synthetic preservatives.

Reviewer 3 Report
This study evaluated the antioxidant effects of larges and cherry tomato byproducts (LTB and CTB) on lipid oxidation and production of functional sausage during refrigerated storage for 20 days. The content of total, individual polyphenols and lycopene of large tomatoes (LT) and cherry tomatoes (CT) samples, as well as their byproducts, was initially evaluated. For the same dose of tomatoes processing byproduct, it was noted a stronger inhibitory effect against lipid oxidation in the case of smoked and scalded sausages compared to smoked and dried.The manuscript is not yet ready for publication. The design of the experiment is relatively simple. It needs a lot of modification. Detailed comments are as follows:
1. Lines 26-28, what do DCTB and DLTB stand for? Please explain.
2.In lines 35-36,“Sausages are a category of meat preparations widely consumed in many countries and are formed from a mixture of proteins and fats from meat with additives.This sentence requires a reference. And it should not be a separate segment.
3.In lines 102-104,"All the chemicals and reagents used were of analytical grade and were procured from Sigma-Aldrich (St Louis MO, USA) and Chimreactiv (Bucharest, Romania) "this sentence does not have a full stop, please correct.
4."2.5. Proximate composition of Sausages", "2.6. Oxidative Stability Assessment", "2.5.1. Determination of oxidant Value There is something wrong with the order of (PV) ", please modify it.
5. The explanation of the formula in the manuscript is relatively simple. Please explain the parameters in the formula in detail.
6.Line 110-111 "After drying, The Samples were grounded with a Grinder (GM 2000, Grindomix) until samples were turned into a fine powder. "What was the size of the powder?
7. Figures in the manuscript should be beautified. Please unify the number of decimal places reserved in the table.
8. Lines307-309 "Significant values have also been reported for gallic acid, Caffeic acid, Caffeic acid, Caffeic acid, Caffeic acid, Caffeic acid, Caffeic acid, Caffeic acid. epicatechin, resveratrol and quercetin, and p-coumaric acid, ferulic acid, Rosmarinic acid and kaempferol were identified at much lower levels. "This sentence requires a reference.
9. "3.2. The proximate composition of learning", "3.5. Fightoxidative stability assessment",there are problems in the ordering here, please modify it.
10. Please enhance the discussion section in "3. Results".
